# Extremely Rare Case of Successful Treatment of Foot Ulcer Associated with Evans’ Syndrome and Buerger’s Disease

**DOI:** 10.3390/medicina60071147

**Published:** 2024-07-16

**Authors:** Ha-Jong Nam, Se-Young Kim, Je-Yeon Byeon, Hwan-Jun Choi

**Affiliations:** 1Department of Plastic and Reconstructive Surgery, Soonchunhyang University Gumi Hospital, Gumi 39371, Republic of Korea; 125039@schmc.ac.kr (H.-J.N.); 111459@schmc.ac.kr (S.-Y.K.); 2Department of Plastic and Reconstructive Surgery, Soonchunhyang University Cheonan Hospital, Cheonan 31538, Republic of Korea; 115954@schmc.ac.kr

**Keywords:** Evans syndrome, thromboangiitis, obliterans, thrombosis, autoimmune disease, case reports

## Abstract

Evans Syndrome (ES) is a rare autoimmune disorder characterized by the simultaneous occurrence of immune thrombocytopenia (ITP) and autoimmune hemolytic anemia (AIHA). Thrombotic complications in ES patients are uncommon, particularly involving Buerger’s Disease (BD). We report a case of a 49-year-old male with ES and a history of diabetes and heavy smoking, presenting with a necrotic wound on his right great toe. Diagnostic evaluations revealed severe stenosis and thrombosis in the lower limb arteries, diagnosed as BD. The patient underwent successful popliteal–tibioperoneal artery bypass surgery and the subsequent disarticulation and revision of the distal phalanx, followed by the application of an acellular dermal matrix (ADM) to promote healing. Post-surgery, the patient showed significant improvement in blood flow and complete epithelialization without complications. This case highlights the importance of a multidisciplinary approach to managing complex wounds in ES patients, suggesting potential treatment pathways for future cases involving BD.

## 1. Introduction

Evans syndrome (ES), first described by Evans in 1951, is defined as the simultaneous or sequential occurrence of immune thrombocytopenia (ITP) and warm autoimmune hemolytic anemia (AIHA) [1]. Studies performed over the past 40 years have revealed that ES has an annual incidence of only 1.8 cases per million people and a prevalence of 21.3 cases per million [2]. Because of its rarity, comparative clinical trials focusing on treatment methodologies are limited, and most current treatment recommendations are derived from those established for ITP and AIHA. The current treatment for BD typically involves smoking cessation, medication for improving blood flow, and surgical interventions if necessary, while treatment for ES includes steroids, intravenous immunoglobulin, and immunosuppressive agents. Additionally, studies examining the involvement of plastic surgery departments in treating wounds in patients with ES are rare [3,4].

Here, we report a novel case of a patient typically managed for ES who concurrently developed Buerger’s disease (BD), also known as thromboangiitis obliterans. BD is a rare inflammatory condition primarily affecting the blood vessels of the arms and legs. This condition is characterized by inflammation and clotting in small- and medium-sized vessels, which reduces blood flow to the affected areas [5]. Reduced blood flow can lead to pain, numbness, and swelling within angiosomes; patients may exhibit a bluish appearance at the extremities because of impaired circulation. Skin necrosis or ulceration can occur in severe cases [6,7].

Acellular Dermal Matrix (ADM) is a biocompatible scaffold derived from human or animal dermis, which has been processed to remove all cells while preserving the extracellular matrix. This matrix serves as a framework for cellular ingrowth and revascularization, promoting wound healing and tissue regeneration. Recent advancements in ADM technology have expanded its applications in various surgical fields, including plastic and reconstructive surgery [8,9]. The use of ADM facilitated secondary intention healing, resulting in significant wound improvement without complications, underscoring its potential utility in complex wound management scenarios associated with ES [10].

Although a direct correlation between ES and thrombogenic diseases such as BD has not been conclusively established, recent studies have indicated that autoimmune hemolytic anemia, a component of ES, is associated with a heightened risk of venous thromboembolism [11]. This case report reviews the association between ES and thrombotic complications, emphasizing the clinical diversity and complexity of ES. Our case of the successful management of BD using a multidisciplinary approach demonstrates the importance of tailored management strategies and the treatment of associated complications in ES, specifically through the application of ADM.

## 2. Case Presentation

### 2.1. Initial Clinical Findings

A 49-year-old male, diagnosed with ES 3 years prior at our institution, presented with an open wound in his right great toe. Initial examination revealed symptoms of blue toe syndrome along with the partial exposure of the distal phalanx (Figure 1).

### 2.2. Medical and Treatment History

The patient was a smoker with a 20-pack-year history and had been under continuous medical supervision for ES. The patient had a history of type 2 diabetes mellitus, diagnosed 5 years prior, that was well-controlled with metformin, and had an HbA1c level of 6.5%. No other significant conditions were identified. In 2018, the patient presented to our emergency department with persistent epistaxis, gingival bleeding, and petechial hemorrhage. These symptoms prompted his admission to the hematology–oncology department, where he began treatment with steroid pulse therapy. Multiple interventions were implemented following disease progression, characterized by a continuous decline in the platelet count. These interventions included repeated doses of intravenous immunoglobulin and platelet transfusions, ultimately necessitating a splenectomy in 2019 (Table 1).

### 2.3. Diagnostic Evaluation

In April 2022, the patient returned to our facility one week after experiencing direct trauma to his toe, which led to necrosis of the distal phalanx. Diagnostic evaluations revealed an ankle–brachial index (ABI) of 0.85 on the right side and 1.3 on the left side, suggesting potential vascular occlusion. Further assessment using transcutaneous oxygen pressure showed measurements of 15 and 19 mmHg on the right and left sides, respectively. Detailed imaging studies, including lower-extremity angio-computed tomography (CT), identified thrombosis in both the popliteal and distal anterior tibial arteries (Figure 2A). The imaging results, along with the characteristic corkscrew pattern observed on a subsequent angiography, led to a diagnosis of BD (Figure 2B). Additionally, 3-dimensional CT showed no specific findings in the right big toe (Figure 2C). However, white blood cell single-photon emission CT imaging showed uptake in the distal phalanx consistent with signs of osteomyelitis (Figure 2D).

### 2.4. Intervention and Outcomes at Vascular Surgery Department

Twelve days after admission, an interdisciplinary approach involving both the cardiology and vascular surgery teams was employed to address the patient’s severe vascular complications. The initial attempt at angioplasty was hindered by the pronounced stenosis of the popliteal artery. Consequently, a decision was made to perform a right popliteal–tibioperoneal artery bypass using the great saphenous vein to restore adequate blood flow. Postoperative imaging using follow-up CT showed significant improvements in popliteal flow. Furthermore, ABI measurements showed a marked improvement, with readings of 0.97 on the right and 1.25 on the left, indicating successful vascular restoration. Three days after bypass surgery, the great toe displayed mild induration but remained stable and well-demarcated, indicating that the surgical intervention was successful (Figure 3).

### 2.5. Surgical Procedures at Plastic Surgery Department

On day 21 of hospitalization, the disarticulation and revision of the distal phalanx was performed under local anesthesia to address ongoing issues in the demarcated area. During the disarticulation procedure, we removed only the distal phalanx, which showed prominent signs of osteomyelitis, including the cartilage surface. Despite the progression of tip necrosis, debridement revealed a healthy articular surface on the mid phalanx (Figure 4A). This favorable outcome enabled the application of an acellular dermal matrix (SureDerm^®^, HansBiomed Corp, Seoul, Republic of Korea), which was used to promote healing and tissue regeneration at the injury site (Figure 4B,C). Culture specimens were obtained from the affected distal phalanx, and initially, intravenous broad-spectrum antibiotics, specifically ampicillin and sulbactam, were administered. Upon identifying Methicillin-Resistant *Staphylococcus Aureus* (MRSA) in the culture results, we consulted with the infectious disease team and switched the antibiotic regimen to intravenous vancomycin to effectively target the MRSA infection.

### 2.6. Discharge and Follow-Up

The patient was discharged after two weeks of dressing changes, during which initial epithelialization commenced from a 20% margin (Figure 5A). Outpatient follow-up visits were scheduled biweekly to monitor the patient’s progress. Full epithelialization was achieved within two weeks of discharge. No complications or adverse effects were observed during the six-month follow-up period (Figure 5B,C).

### 2.7. Informed Consent

The patient provided written informed consent for the publication and use of his images.

## 3. Discussion

ES is a rare and complex autoimmune disorder characterized by the simultaneous or sequential occurrence of warm AIHA and ITP [1,12]. This condition often presents alongside autoimmune neutropenia, contributing to a variety of clinical symptoms due to anemia, thrombocytopenia, and leukopenia. Anemia associated with ES typically results from warm IgG autoantibodies, specifically IgA, but not cold agglutinins. Autoimmune neutropenia can also manifest as a symptom of ES, occurring in 15% adults and 20% of children [12].

ES symptoms can resemble those of leukemia and lymphoma, necessitating a differential diagnosis to exclude these conditions. Hemolysis can lead to jaundice, dark-brown urine, pallor, weakness, fatigue, and dyspnea. Thrombocytopenia may cause increased bruising and petechiae, which appear as small vessels bleeding into the skin, making blood vessels appear more prominent. Additionally, patients may exhibit bleeding tendencies such as epistaxis and menorrhagia. A low neutrophil count increases the risk of fever, oral ulcers, and bacterial infections [4].

The diagnosis of ES is based on characteristic symptoms, the patient’s history, a thorough clinical evaluation, and various specialized tests. Although there is no definitive test for ES, it is often diagnosed by excluding other potential conditions [3,4]. Specifically, ES can be diagnosed in the presence of both AIHA (positive direct Coombs test) and ITP in the same patient, regardless of whether these conditions occur simultaneously or sequentially [3,12]. In this case, the diagnosis was also based on symptoms and signs, with a positive direct Coombs test.

The first-line treatments for ES typically involve steroids such as prednisone and prednisolone. In severe cases of immune thrombocytopenic purpura, intravenous immunoglobulin may be administered as life-saving therapy. The second-line treatments for refractory ES include rituximab, mycophenolate mofetil, cyclosporine, vincristine, azathioprine, sirolimus, and thrombopoietin receptor agonists [3,12]. The patient in this case was also treated with blood and platelet transfusions and primarily treated with steroids. Our therapeutic approach was consistent with those reported in the literature, starting with intravenous steroids and later including splenectomy and intravenous immunoglobulin when persistent thrombocytopenia and steroid resistance were observed. Steroids primarily act by removing macrophages, which destroy red blood cells and platelets. Concentrated IgG from human plasma donors blocks Fcγ receptors on macrophages, although this treatment remains controversial [13]. Splenectomy is effective for treating thrombocytopenia; however, the potential side effects must be considered [14].

Reports of thrombotic complications associated with ES are rare. Neurological complications, such as cerebral thrombosis or hemorrhage, can occur, leading to severe headaches, language difficulties, or paralysis [4]. Pulmonary embolism should be suspected in patients showing respiratory symptoms, such as difficulty breathing and chest pain. Thrombotic complications can be a serious issue in patients with ES because of their decreased platelet counts and anemia, which can reduce their blood clotting abilities [4]. Therefore, the accurate diagnosis and treatment of thrombotic complications in patients with ES are crucial. Diagnostic tests for thrombotic complications in patients with ES may include blood tests such as D-dimer and coagulation studies, whereas imaging studies such as ultrasound and CT can help specify the location and severity of thrombi. Anticoagulation therapy and thrombolytic agents may be considered to prevent and manage thrombotic complications [4].

Because of the very low prevalence of ES, research on the association between ES and thrombotic complications is lacking. However, previous studies reported an increased likelihood of thrombosis as a complication of ITP and AIHA [15]. To our knowledge, no cases of lower-limb thrombotic complications in patients with ES have been reported to date. In our case, BD was accompanied by severe stenosis of the popliteal artery, as observed in imaging (Figure 2A). BD primarily involves inflammation and thrombus formation in the small-to medium-sized vessels of the lower limbs, which can lead to the narrowing and occlusion of the vessels, cutting off of the blood supply to related tissues, and symptoms such as pain and blue toe syndrome with tissue damage [5]. Typical corkscrew patterns were observed during the angiography (Figure 2B). Therefore, appropriate radiological assessment and intervention planning considering BD are necessary in cases of lower-limb ulcers in patients with ES.

Our patient was previously treated for ES and had undergone splenectomy and steroid therapy; he also had a history of heavy smoking and diabetes. The patient showed improved blood flow and the verified recovery of the ABI following vascular surgery and bypass surgery. Thus, a multidisciplinary approach should be considered for managing patients with ES and associated BD. The patient achieved successful epithelialization following debridement of the necrotic tissue, including the distal phalanx. During this process, an acellular dermal matrix was applied to promote healing, minimize damage to the surrounding tissues, and facilitate secondary intention healing. This method also provides a scaffold that supports cellular infiltration and tissue regeneration.

ADM is a biological scaffold derived from human or animal dermis, which has been processed to remove all cells while preserving the extracellular matrix. This matrix serves as a framework for cellular ingrowth and neovascularization, promoting wound healing and tissue regeneration. Recent studies have highlighted the efficacy of ADM, particularly when used as a co-graft with split-thickness skin grafts (STSG), in enhancing wound healing outcomes in complex cases [11]. In this case, ADM was chosen over traditional grafting methods due to the patient’s extensive medical history, including ES, diabetes, and heavy smoking, which increased the risk of graft failure and complications. ADM provided a more suitable option as it supports cellular infiltration and tissue regeneration while minimizing the potential for immune rejection and other complications [12,13]. The patient recovered completely without signs of infection or other complications, underscoring the effectiveness of this approach in managing complex wounds associated with ES and BD. The decision to use ADM was validated by the positive outcomes observed, demonstrating its potential as a valuable tool in reconstructive surgery, particularly for patients with challenging medical histories.

The primary limitation of this study is its reliance on a single patient. Thus, the reproducibility and generalizability of our results must be further analyzed before universal guidelines are recommended. However, based on the successful recovery of this patient, we suggest directions for treatment and diagnostic plans for future cases of ES with BD. Furthermore, BD is a potential complication of ES, underscoring the need for precise diagnostic strategies and tailored treatment plans for patients with ES and highlighting the critical role of a multidisciplinary approach to managing the complexities of this disorder.

## 4. Conclusions

This case highlights the critical importance of a multidisciplinary approach to managing complex wounds in patients with ES and BD. The successful treatment of our patient underscores the potential use of ADM in promoting healing and suggests promising pathways for future cases involving similar conditions. Further research is warranted to explore these findings in larger cohorts.

## Figures and Tables

**Figure 1 medicina-60-01147-f001:**
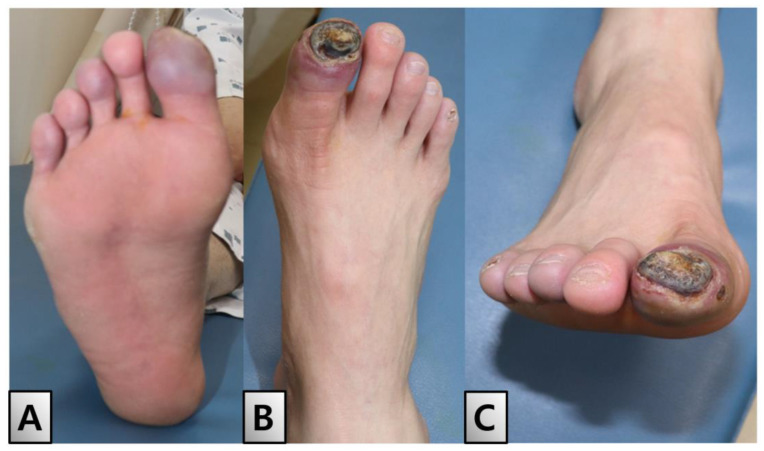
Initial findings. The entire right foot was chilled, and the right great toe had a bluish color with the formation of a 2 × 1.5 cm sized eschar on the toe tip, with the bone exposed. (**A**) Plantar view. (**B**) Dorsal view. (**C**) Anterior view.

**Figure 2 medicina-60-01147-f002:**
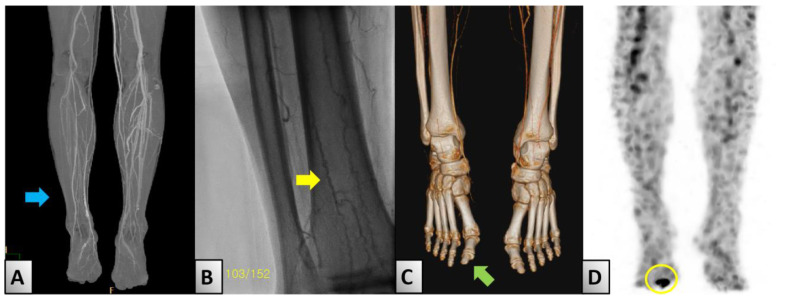
Radiologic findings. (**A**) Lower-extremity computed tomography shows severe stenosis in the right popliteal artery (blue arrow). (**B**) Lower-extremity angiography shows a ‘corkscrew appearance’ (yellow arrow), which is characteristic of Buerger’s disease. (**C**) 3-dimensional CT shows no specific findings in the right big toe (green arrow). (**D**) White blood cell single-photon emission computed tomography showed osteomyelitis in the right big toe (yellow circle). Abbreviations: CT; computed tomography.

**Figure 3 medicina-60-01147-f003:**
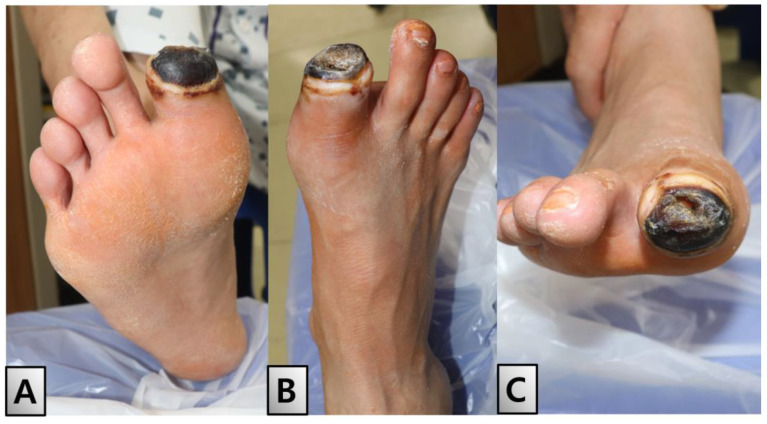
Post-bypass surgery status of the right great toe. Three days following bypass surgery, the right big toe displayed mild induration but remained stable and well-demarcated, indicating the initial success of the surgical intervention. (**A**) Plantar view. (**B**) Dorsal view. (**C**) Anterior view.

**Figure 4 medicina-60-01147-f004:**
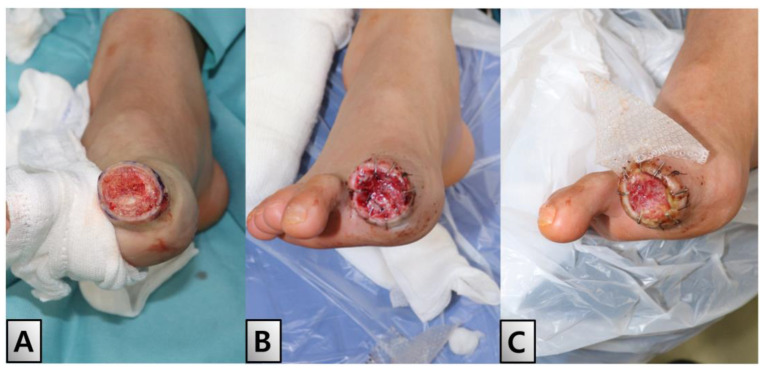
Post-disarticulation and healing process. On day 21 of hospitalization, a disarticulation and revision of the distal phalanx was performed under local anesthesia. Despite necrosis, debridement revealed a healthy raw surface, allowing for the application of an acellular dermal matrix (SureDerm^®^, HansBiomed Corp, Seoul, Republic of Korea). (**A**) Post-debridement view. (**B**) Application of the dermal matrix. (**C**) Healing with the well-taken matrix in place.

**Figure 5 medicina-60-01147-f005:**
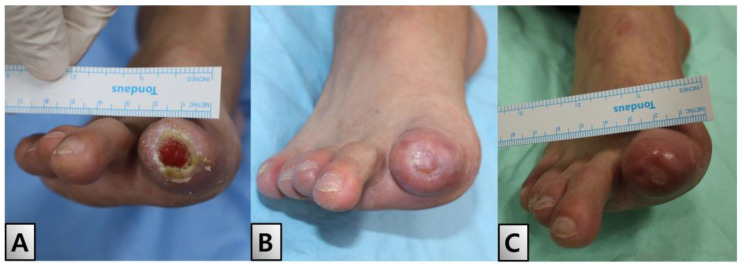
Discharge and follow-up. (**A**) Patient was discharged after two weeks, during which initial epithelialization commenced from a 20% margin. Outpatient follow-up visits were scheduled bi-weekly to monitor progress. (**B**) Full epithelialization achieved two weeks after discharge. (**C**) Condition six months post-discharge. Throughout the follow-up period, no complications or adverse effects were reported.

**Table 1 medicina-60-01147-t001:** Comprehensive medical timeline of the patient with Evans syndrome.

Date	Event	Details
Underlying Disease	Diagnosed with type 2 diabetes mellitus	Well controlled: HbA1c level of 6.5%
11 February 2018	Visits emergency room first timeFirst diagnosis of Evans Syndrome	Presented with epistaxis, gingival bleeding, and petechial hemorrhages. Antibody screening test: positive. Direct Coombs test: 4 positive.
2018–2019	Continuous treatment	Multiple interventions including repeated doses of intravenous immunoglobulin and platelet transfusions.
3 August 2019–13 August 2019	Admission for laparoscopic splenectomy	Surgery performed to manage Evans Syndrome effectively.
18 April 2022	Admission by Plastic Surgery Department	For treatment of wound in right big toe.
21 April 2022	Lower-extremity ct	Focal thrombus identified in right popliteal artery and distal ATA.
4 May 2022	Right femoral arteriography	Severe stenosis in right popliteal artery; chronic total occlusions in right ATA, right posterior tibial artery (PTA), and right peroneal artery Corkscrew findings suggestive of Buerger’s Disease.
12 May 2022	Right popliteal–tibioperoneal artery bypass surgery with GSV.	Surgical intervention to bypass identified thrombus, using GSV.

Abbreviations: ATA; anterior tibial artery, PTA; posterior tibial artery, GSV; great saphenous vein. The patient was admitted to the plastic surgery department for the treatment of a wound on the right big toe.

## Data Availability

The data presented in this study are available upon reasonable request from the corresponding author.

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
