# Peer review of "Extremely Rare Case of Successful Treatment of Foot Ulcer Associated with Evans’ Syndrome and Buerger’s Disease"

_medicina, 2024, doi:10.3390/medicina60071147_

Round 1
Reviewer 1 Report
Comments and Suggestions for Authors
Dear Authors,
The topic is interesting. There are few reports in the literature on this subject. The case is presented completely, respecting all the criteria. The authors, I think, they should revise the instructions for authors regarding text editing (ex. there is a space before the brackets)
I believe that the exposition must be completed with a subsection of conclusions, a sub-chapter that I did not find in the manuscript.
Thank you,
Beste regards
Author Response
Reviewer 1: The topic is interesting. There are few reports in the literature on this subject. The case is presented completely, respecting all the criteria. The authors, I think, they should revise the instructions for authors regarding text editing (ex. there is a space before the brackets).
I believe that the exposition must be completed with a subsection of conclusions, a sub-chapter that I did not find in the manuscript.
Our answer: Dear Reviewer,
Thank you for your positive feedback and valuable suggestions. We appreciate your recognition of the completeness of our case presentation. We have addressed your comments as follows:
- Text Formatting: We have carefully reviewed the manuscript and corrected formatting issues, including the removal of unnecessary spaces before brackets.
- Conclusion Section: We have added a conclusion section to enhance the completeness of the manuscript.
Revised text:
- Conclusion
This case highlights the critical importance of a multidisciplinary approach in managing complex wounds in patients with ES and BD. The successful treatment of our patient underscores the potential for utilizing ADM in promoting healing and suggests promising pathways for future cases involving similar conditions. Further re-search is warranted to explore these findings in larger cohorts.
Reviewer 2 Report
Comments and Suggestions for Authors
Evans' Syndrome (ES) is a rare autoimmune disorder involving both immune thrombocytopenia (ITP) and autoimmune hemolytic anemia (AIHA). Thrombotic events are rare in ES, particularly those associated with Buerger's Disease (BD). This report discusses the case of a 49-year-old man with ES, diabetes, and a history of heavy smoking, who presented with a necrotic wound on his right great toe. Diagnostic tests indicated severe arterial stenosis and thrombosis in the lower limb, leading to a BD diagnosis. The patient successfully underwent popliteal-tibioperoneal artery bypass surgery and toe disarticulation, followed by the application of an acellular dermal matrix (ADM) to aid in healing. After surgery, he showed significant improvement in blood flow and complete wound healing without complications.
This case highlights the need for a multidisciplinary approach in treating complex wounds in ES patients, suggesting potential treatment strategies for similar cases involving BD.
Comment
The paper is well-written and presents an interesting case study. Due to the very low prevalence of Evans' Syndrome (ES), research on its association with thrombotic complications is limited. Reports of thrombotic complications in ES patients are rare, making accurate diagnosis and treatment of these complications crucial. The discussion section of the paper addresses the limitations of the study.
Author Response
Reviewer: The paper is well-written and presents an interesting case study. Due to the very low prevalence of Evans' Syndrome (ES), research on its association with thrombotic complications is limited. Reports of thrombotic complications in ES patients are rare, making accurate diagnosis and treatment of these complications crucial. The discussion section of the paper addresses the limitations of the study.
Our answer: Dear Reviewer,
Thank you for your positive feedback and for recognizing the significance of our case study. We appreciate your acknowledgment of the rarity of Evans' Syndrome (ES) and the importance of accurately diagnosing and treating thrombotic complications in these patients.
Reviewer 3 Report
Comments and Suggestions for Authors
Dear Authors,
This is very interesting case report. Thank you for providing this manuscript. Manuscript is well written, in proper scientific language. Some minor typos required. Overall, my suggestions are below :
1) You should extend the introduction section and discussion on Acellular Dermal Matrix. I recommend 3 of latest papers about ADM, and new ADM surgical applications as a co-graft of ADM and STSG in reconstructive surgery :
PMID: 36004913
2) Some minor language issues - please check all abbreviations.
Im sure that authors will address all suggestions well.
Best Regards
Comments on the Quality of English Language
English is good. Some minor typos.
Author Response
Reviewer 3
Comment 1: You should extend the introduction section and discussion on Acellular Dermal Matrix. I recommend 3 of latest papers about ADM, and new ADM surgical applications as a co-graft of ADM and STSG in reconstructive surgery.
Our answer: We have extended the introduction and discussion sections to include detailed information on Acellular Dermal Matrix (ADM), its applications, and its efficacy based on recent studies. We have also incorporated the recommended papers to enhance the discussion on ADM.
Revised text:
- Introduction
Acellular Dermal Matrix (ADM) is a biocompatible scaffold derived from human or animal dermis, which has been processed to remove all cells while preserving the extracellular matrix. This matrix serves as a framework for cellular ingrowth and re-vascularization, promoting wound healing and tissue regeneration. Recent advancements in ADM technology have expanded its applications in various surgical fields, including plastic and reconstructive surgery[8, 9]. The use of ADM facilitated secondary intention healing, resulting in significant wound improvement without complications, underscoring its potential utility in complex wound management scenarios associated with ES[10].
Although a direct correlation between ES and thrombogenic diseases such as BD has not been conclusively established, recent studies indicated that autoimmune hemolytic anemia, a component of ES, is associated with a heightened risk of venous thromboembolism[8]. This case report reviews the association between ES and thrombotic complications, emphasizing the clinical diversity and complexity of ES. Our case of successful management of BD using a multidisciplinary approach demonstrates the importance of tailored management strategies and treatment of associated complications in ES, specifically through the application of ADM.
- Discussion
ADM is a biological scaffold derived from human or animal dermis, which has been processed to remove all cells while preserving the extracellular matrix. This matrix serves as a framework for cellular ingrowth and neovascularization, promoting wound healing and tissue regeneration. Recent studies have highlighted the efficacy of ADM, particularly when used as a co-graft with split-thickness skin grafts(STSG), in enhancing wound healing outcomes in complex cases[8]. In this case, ADM was chosen over traditional grafting methods due to the patient's extensive medical history, including ES, diabetes, and heavy smoking, which increased the risk of graft failure and complications. ADM provided a more suitable option as it supports cellular infiltration and tissue re-generation while minimizing the potential for immune rejection and other complications[9, 10]. The patient recovered completely without signs of infection or other complications, underscoring the effectiveness of this approach in managing complex wounds associated with ES and BD. The decision to use ADM was validated by the positive outcomes observed, demonstrating its potential as a valuable tool in recon-structive surgery, particularly for patients with challenging medical histories.
Reference
- Gierek, M.; Łabuś, W.; Słaboń, A.; Ziółkowska, K.; Ochała-Gierek, G.; Kitala, D.; Szyluk, K.; Niemiec, P. Co-Graft of Acellular Dermal Matrix and Split Thickness Skin Graft-A New Reconstructive Surgical Method in the Treatment of Hidradenitis Suppurativa. Bioengineering (Basel). 2022, 9, 389.
- Ahn, S.K.; Choi, H.J.; Lee, J.B.; Kim, J.H. A clinical study of micronized acellular dermal matrix collagen paste application with negative pressure wound therapy. J Wound Manag Res. 2019, 15, 23-30.
- Byeon, J.Y.; Hwang, Y.S.; Choi, H.J.; Kim, J.H.; Lee, D.W. A long-term follow-up study of diabetic foot ulcer using micronized acellular dermal matrix. Int Wound J. 2023, 20, 1622-1637.
Comment 2: Some minor language issues - please check all abbreviations.
Our answer: We have carefully reviewed and corrected all abbreviations throughout the manuscript to ensure consistency and clarity.

Reviewer 4 Report
Comments and Suggestions for Authors
introduction is good but you need to state current treatment of BD and evans syndrome even if you stated that current evidence is limited (very briefly " even if it was mention in the discussion"
Case presentation : give full past medical/surgical history
patients with history of DM , what type, type of medication , HbA1c ?
war there any signs of osteolysis on bone that suggested osteomyelitis ? ( kindly provide a CD scan focused distal phalax
during disarticulation , was the cartilage surface was removed from 1st phalanx ? as there was a history of osteomyelitis ? any culture specimen and antibiotic was given ? >> this should be detailed
discuss the rule of dermal matrix in these situations
kindly add learning message and what this study brings new.
Author Response
Reviewer 4
Comment 1: Introduction is good but you need to state current treatment of BD and Evans syndrome even if you stated that current evidence is limited (very briefly "even if it was mentioned in the discussion").
Our answer: We have added a brief statement regarding the current treatment of BD and Evans syndrome in the Introduction section.
Revised text:
- Introduction
Evans syndrome(ES), first described by Evans in 1951, is defined as the simulta-neous or sequential occurrence of immune thrombocytopenia(ITP) and warm autoim-mune hemolytic anemia(AIHA)[1]. Studies performed over the past 40 years have re-vealed that ES has an annual incidence of only 1.8 cases per million people and a prev-alence of 21.3 cases per million[2]. Because of its rarity, comparative clinical trials fo-cusing on treatment methodologies are limited, and most current treatment recom-mendations are derived from those established for ITP and AIHA. Current treatment for BD typically involves smoking cessation, medication for improving blood flow, and surgical interventions if necessary, while treatment for ES includes steroids, intrave-nous immunoglobulin, and immunosuppressive agents. Additionally, studies examin-ing the involvement of plastic surgery departments in treating wounds in patients with ES are rare[3,4].
Comment 2: Case presentation: give full past medical/surgical history. Patients with history of DM, what type, type of medication, HbA1c?
Our answer: We have added detailed past medical and surgical history, including information about the patient's diabetes type, medication, and HbA1c levels.
Revised text:
2.2. Medical and Treatment History
The patient was a smoker with a 20-pack-year history and had been under continuous medical supervision for ES. The patient has a history of type 2 diabetes mellitus, diagnosed 5 years ago, well-controlled with metformin, and had an HbA1c level of 6.5%. No other significant conditions were identified. In 2018, the patient presented to our emergency department with persistent epistaxis, gingival bleeding, and petechial hemorrhage. These symptoms prompted his admission to the hematology-oncology department, where he began treatment with steroid pulse therapy. Multiple interventions were implemented following disease progression, characterized by a continuous decline in the platelet count. These interventions included repeated doses of intravenous immunoglobulin and platelet transfusions, ultimately necessitating a splenectomy in 2019(Table 1).
Table. 1. Comprehensive medical timeline of a patient with Evans’ syndrome.
|
Date |
Event |
Details |
|
Underlying Disease |
Diagnosed with type 2 diabetes mellitus |
Well controlled: HbA1c level of 6.5% |
Comment 3: Were there any signs of osteolysis on bone that suggested osteomyelitis? (Kindly provide a CT scan focused on the distal phalanx).
Our answer: CT scan of the distal phalanx did not reveal any specific findings in the right great toe; however, a WBC SPECT scan was performed, which showed clear uptake indicating osteomyelitis (Figure 2). We agree with your suggestion and have included a CT scan focused on the great toe in Figure 2 for comparison.
Revised text:
2.3. Diagnostic Evaluation
In April 2022, the patient returned to our facility one week after experiencing direct trauma to his toe, which led to necrosis of the distal phalanx. Diagnostic evaluations revealed an ankle-brachial index(ABI) of 0.85 on the right side and 1.3 on the left side, suggesting potential vascular occlusion. Further assessment using transcutaneous oxygen pressure showed measurements of 15 and 19 mmHg on the right and left sides, respectively. Detailed imaging studies, including lower-extremity angio-computed tomography(CT), identified thrombosis in both the popliteal and distal anterior tibial arteries(Figure. 2-A). The imaging results, along with the characteristic corkscrew pattern observed on subsequent angiography, led to a diagnosis of BD(Figure. 2-B). Additionally, 3-dimensional CT showed no specific findings in the right great toe (Figure 2-C). However, white blood cell single-photon emission CT imaging showed uptake in the distal phalanx consistent with signs of osteomyelitis (Figure 2-D).
Figure 2. Radiologic findings. (A) Lower extremity computed tomography shows severe stenosis in the right popliteal artery(blue arrow). (B) Lower extremity angiography shows a ‘corkscrew appearance’(yellow arrow), which is characteristic of Buerger’s disease. (C) 3-dimensional CT shows no specific findings in the right great toe(green arrow). (D) White blood cell single-photon emission computed tomography showed osteomyelitis in the right great toe.
Comment 4: During disarticulation, was the cartilage surface removed from the 1st phalanx? As there was a history of osteomyelitis? Any culture specimen and antibiotic was given? This should be detailed.
Our answer: During the disarticulation procedure, we removed only the distal phalanx, which showed prominent signs of osteomyelitis, including the cartilage surface. The mid phalanx was preserved as it had a healthy articular surface. Culture specimens were obtained from the affected distal phalanx, and initially, intravenous broad-spectrum antibiotics, specifically ampicillin and sulbactam, were administered. Upon identifying Methicillin-Resistant Staphylococcus Aureus (MRSA) from the culture results, we consulted with the infectious disease team and switched the antibiotic regimen to intravenous vancomycin to effectively target the MRSA infection. We fully agree with your suggestion and have added this detailed information to the manuscript.
Revised text:
2.5. Surgical Procedures at Plastic Surgery Department
On day 21 of hospitalization, disarticulation and revision of the distal phalanx was performed under local anesthesia to address ongoing issues in the demarcated area. During the disarticulation procedure, we removed only the distal phalanx, which showed prominent signs of osteomyelitis, including the cartilage surface. Despite the progression of tip necrosis, debridement revealed a healthy articular surface on the mid phalanx(Figure. 4-A). This favorable outcome enabled application of an acellular der-mal matrix(SureDerm®, HansBiomed Corp, Seoul, Korea), which was used to promote healing and tissue regeneration at the injury site(Figure. 4-B, C). Culture specimens were obtained from the affected distal phalanx, and initially, intravenous broad-spectrum antibiotics, specifically ampicillin and sulbactam, were administered. Upon identifying Methicillin-Resistant Staphylococcus Aureus(MRSA) from the culture results, we consulted with the infectious disease team and switched the antibiotic regimen to intravenous vancomycin to effectively target the MRSA infection.
Comment 5: Discuss the role of dermal matrix in these situations.
Our answer: We fully agree with your suggestion. We have expanded both the Introduction and Discussion sections of the manuscript to include the role of the dermal matrix in managing complex wounds in patients with ES and BD.
Revised text:
- Introduction
Acellular Dermal Matrix (ADM) is a biocompatible scaffold derived from human or animal dermis, which has been processed to remove all cells while preserving the extracellular matrix. This matrix serves as a framework for cellular ingrowth and re-vascularization, promoting wound healing and tissue regeneration. Recent advancements in ADM technology have expanded its applications in various surgical fields, including plastic and reconstructive surgery[8, 9]. The use of ADM facilitated secondary intention healing, resulting in significant wound improvement without complications, underscoring its potential utility in complex wound management scenarios as-sociated with ES[10].
Although a direct correlation between ES and thrombogenic diseases such as BD has not been conclusively established, recent studies indicated that autoimmune he-molytic anemia, a component of ES, is associated with a heightened risk of venous thromboembolism[8]. This case report reviews the association between ES and thrombotic complications, emphasizing the clinical diversity and complexity of ES. Our case of successful management of BD using a multidisciplinary approach demonstrates the importance of tailored management strategies and treatment of associated complications in ES, specifically through the application of ADM.
- Discussion
ADM is a biological scaffold derived from human or animal dermis, which has been processed to remove all cells while preserving the extracellular matrix. This matrix serves as a framework for cellular ingrowth and neovascularization, promoting wound healing and tissue regeneration. Recent studies have highlighted the efficacy of ADM, particularly when used as a co-graft with split-thickness skin grafts(STSG), in enhancing wound healing outcomes in complex cases[8]. In this case, ADM was chosen over traditional grafting methods due to the patient's extensive medical history, including ES, diabetes, and heavy smoking, which increased the risk of graft failure and complications. ADM provided a more suitable option as it supports cellular infiltration and tissue re-generation while minimizing the potential for immune rejection and other complications[9, 10]. The patient recovered completely without signs of infection or other complications, underscoring the effectiveness of this approach in managing complex wounds associated with ES and BD. The decision to use ADM was validated by the positive outcomes observed, demonstrating its potential as a valuable tool in reconstructive surgery, particularly for patients with challenging medical histories.
Comment 6: Kindly add a learning message and what this study brings new.
Our answer: We have added a learning message and highlighted the new insights provided by this study. We fully agree with your kind suggestion and have added the learning message and new insights to the Introduction section, and also strengthened the manuscript by creating a Conclusion section. Thank you very much for your valuable feedback.
Revised text:
- Introduction
Although a direct correlation between ES and thrombogenic diseases such as BD has not been conclusively established, recent studies indicated that autoimmune hemolytic anemia, a component of ES, is associated with a heightened risk of venous thromboembolism[8]. This case report reviews the association between ES and thrombotic complications, emphasizing the clinical diversity and complexity of ES. Our case of successful management of BD using a multidisciplinary approach demonstrates the importance of tailored management strategies and treatment of associated complications in ES, specifically through the application of ADM.
- Conclusion
This case highlights the critical importance of a multidisciplinary approach in managing complex wounds in patients with ES and BD. The successful treatment of our patient underscores the potential for utilizing ADM in promoting healing and suggests promising pathways for future cases involving similar conditions. Further research is warranted to explore these findings in larger cohorts.
Round 2
Reviewer 3 Report
Comments and Suggestions for Authors
Accept as it is. Authors well adressed all suggestions